# TöRF: Time-of-Flight Radiance Fields
# for Dynamic Scene View Synthesis

**Benjamin Attal**[*]
Carnegie Mellon University

**Eliot Laidlaw**
Brown University

**Aaron Gokaslan**
Cornell University

**Changil Kim**
Facebook

**Christian Richardt**
University of Bath

**James Tompkin**
Brown University

**Matthew O'Toole**
Carnegie Mellon University

imaging.cs.cmu.edu/torf

## Abstract

Neural networks can represent and accurately reconstruct radiance fields for static 3D scenes (e.g., NeRF). Several works extend these to dynamic scenes captured with monocular video, with promising performance. However, the monocular setting is known to be an under-constrained problem, and so methods rely on data-driven priors for reconstructing dynamic content. We replace these priors with measurements from a time-of-flight (ToF) camera, and introduce a neural representation based on an image formation model for continuous-wave ToF cameras. Instead of working with processed depth maps, we model the raw ToF sensor measurements to improve reconstruction quality and avoid issues with low reflectance regions, multi-path interference, and a sensor's limited unambiguous depth range. We show that this approach improves robustness of dynamic scene reconstruction to erroneous calibration and large motions, and discuss the benefits and limitations of integrating RGB+ToF sensors that are now available on modern smartphones.

## 1 Introduction

Novel-view synthesis (NVS) is a long-standing problem in computer graphics and computer vision, where the objective is to photorealistically render images of a scene from novel viewpoints. Given a number of images taken from different viewpoints, it is possible to infer both the geometry and appearance of a scene, and then use this information to synthesize images at novel camera poses. One of the challenges associated with NVS is that it requires a diverse set of images from different perspectives to accurately represent the scene. This might involve moving a single camera around a static environment [4, 16, 31, 32, 36], or using a large multi-camera system to capture dynamic events from different perspectives [2, 7, 24, 38, 44, 56]. Techniques for dynamic NVS from a monocular video sequence have also demonstrated compelling results, though they suffer from various visual artifacts due to the ill-posed nature of this problem [26, 37, 42, 50, 52]. This requires introducing priors, often deep learned, on the dynamic scene's depth and motion.

In parallel, mobile devices now have camera systems with both color and depth sensors, including Microsoft's Kinect and HoloLens devices, and the front and rear RGBD camera systems in the iPhone and iPad Pro. Depth sensors can use stereo or structured light, or increasingly the more accurate time-of-flight principle for measurements. Although depth sensing technology is more common than ever, many NVS techniques currently do not exploit this additional source of visual information.

To improve NVS performance, we propose TöRF[1], an implicit neural representation for scene appearance that leverages both color and time-of-flight (ToF) images, as depicted in Figure 1. This

---

[*]Correspondence should be addressed to Benjamin Attal: `battal@andrew.cmu.edu`.

[1]TöRF = ToF + NeRF. Pronounced just like 'NeRF' but starts with a 'T'.

35th Conference on Neural Information Processing Systems (NeurIPS 2021).

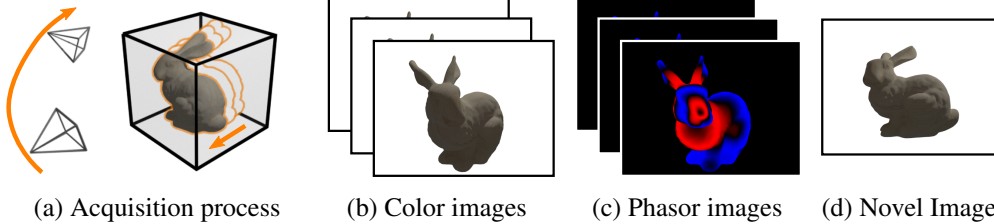

| (a) Acquisition process | (b) Color images | (c) Phasor images | (d) Novel Image |

Figure 1: *Illustration of time-of-flight radiance fields.* **(a)** We move a handheld imaging system around a dynamic scene, capturing **(b)** color images and **(c)** raw phasor images from a continuous-wave time-of-flight (C-ToF) camera. **(d)** Then, we optimize for a continuous neural radiance field of the scene that predicts the captured color and phasor images. This allows novel view synthesis.

reduces the number of images required for static NVS problem settings, compared with just using a color camera. Further, the additional depth information makes the monocular dynamic NVS problem more tractable, as it directly encodes information about the geometry of the scene. Most importantly, rather than using depth directly, we show that using 'raw' ToF data—in the form of phasor images [12] that are normally used to derive depth—is more accurate as it allows the optimization to correctly handle geometry that exceeds the sensor's unambiguous range, objects with low reflectance, and regions affected by multi-path interference, leading to better dynamic scene view synthesis. The contributions of our work include:

- A physically-based neural volume rendering model for raw continuous-wave ToF images;
- A method to optimize a neural radiance field of dynamic scenes with information from color and continuous-wave ToF sensors;
- Quantitative and qualitative evaluation on synthetic and real scenes showing better view synthesis than NeRF [32] for few input views, and than two dynamic scene baselines [26, 52].

## 2   Related Work

While novel-view synthesis (NVS) is a long-standing problem in computer graphics and vision [8, 9, 11, 23], recent developments in neural scene representations for NVS have enabled state-of-the-art results for a wide variety of settings [46, 49]. The common thread across many of these works is to bring learnable elements together with physics-based models and classical rendering processes.

The designs for neural scene representations often build on standard computer graphics data structures, including voxel grids [34, 47], multiplane images (MPIs) [31, 51, 53, 57], multi-sphere images (MSIs) [3, 7], point clouds [30], and implicit functions of scene geometry and appearance [32, 48, 54]. For example, DeepVoxels [47] represent a scene as a discrete volume of embedded features to encode view-dependent effects, and enable wide baselines that may not be possible with other representations; however, the cost associated with this volumetric representation is that the memory requirements scale cubically with resolution. Alternatively, MPIs can be used to encode appearance from a single stereo pair [57]; the key benefit of this representation is the fast rendering speeds (ideal for interactive VR applications), though it performs best for forward-facing scenes.

Implicit neural representations of a scene provide similar flexibility to voxel grids, but circumvent the high memory requirements. These implicit networks therefore have greater capacity to represent the appearance of a scene. For example, scene representation networks (SRNs) [48] encode the geometry in a single neural network, which takes 3D points as input and outputs a feature representation of local scene properties (e.g., surface color or reflectance); rendering an image requires a differentiable ray marching procedure that intersects rays with the implicit volume. Neural radiance fields (NeRFs) [32] encode 5D radiance fields (3D position with 2D viewing direction) to offer higher-fidelity geometry and visual appearance. While these implicit neural representations initially assumed a static scene, recent approaches also demonstrate the ability to perform dynamic NVS from monocular video [26, 37, 42, 50, 52], despite this being a highly ill-posed problem.

Including depth maps has proven beneficial to improve NVS results for a long time [41]. However, surprisingly few NVS methods exploit the availability of depth sensors. One reason is that explicitly reconstructing depth maps for NVS [28, 55] may prove problematic, e.g., for thin structures, depth edges, complex reflectance, or noisy depth. We circumvent these issues by proposing a neural representation that models raw ToF data for better view synthesis for both static and dynamic scenes.

Table 1: Mathematical symbol legend for the following equations and explanations.

| Symbol | Units | Description |
|---|---|---|
| $\mathbf{x}$ | | A point $\in \mathbb{R}^3$. |
| $\boldsymbol{\omega}$ | | A direction; unit vector $\in \mathbb{S}^2$. |
| $\mathbf{n}(\mathbf{x})$ | | A normal; a direction perpendicular to a surface at point $\mathbf{x}$. |
| $\mathbf{x}_t$ | | A point $t$ units along a direction $\boldsymbol{\omega}$, $\mathbf{x}_t = \mathbf{x} + \boldsymbol{\omega}t$. |
| $\boldsymbol{\omega}_\mathrm{i}$ | | A direction incoming to a point. |
| $\boldsymbol{\omega}_\mathrm{o}$ | | A direction outgoing from a point. |
| $L(\mathbf{x}, \boldsymbol{\omega})$ or $L_\mathrm{RGB}$ | $\mathrm{W \cdot sr^{-1} \cdot m^{-2}}$ | Radiance measured by a camera at point $\mathbf{x}$ in direction $\boldsymbol{\omega}$. |
| $L_\mathrm{ToF}(\mathbf{x}, \boldsymbol{\omega})$ | $\mathrm{W \cdot sr^{-1} \cdot m^{-2}}$ | Phasor radiance measured by a C-ToF camera. |
| $L_\mathrm{i}(\mathbf{x}, \boldsymbol{\omega})$ | $\mathrm{W \cdot sr^{-1} \cdot m^{-2}}$ | Incident radiance to a point from a direction. |
| $L_\mathrm{s}(\mathbf{x}, \boldsymbol{\omega})$ | $\mathrm{W \cdot sr^{-1} \cdot m^{-2}}$ | Reflected radiance scattered from a point in a direction. |
| $I$ | $\mathrm{W \cdot sr^{-1}}$ | Radiant intensity of a point light source. |
| $I_\mathrm{s}(\mathbf{x}, \boldsymbol{\omega})$ | $\mathrm{W \cdot sr^{-1}}$ | Reflected radiant intensity scattered from a point $\mathbf{x}$ in direction $\boldsymbol{\omega}$ due to a light source collocated with the camera. |
| $\sigma(\mathbf{x})$ | $\mathrm{m^{-1}}$ | Density function at a point. |
| $T(\mathbf{x}, \mathbf{x}_t)$ | *unitless* | Transmittance function, i.e., accumulated density. |
| $\hat{T}(\mathbf{x}, \mathbf{x}_k)$ | *unitless* | Discrete approximation of the transmittance function. |
| $f_\mathrm{p}(\mathbf{x}, \boldsymbol{\omega}_\mathrm{i}, \boldsymbol{\omega}_\mathrm{o}, \mathbf{n}(\mathbf{x}))$ | $\mathrm{sr^{-1}}$ | Scattering phase function. |
| $W(d)$ | *unitless* | Importance function for light path of length $d$. |

## 3   Neural Volume Rendering of ToF images

A neural radiance field (NeRF) [32] is a neural network optimized to predict a set of input images. Assuming a static scene, the neural network $F_{\boldsymbol{\theta}} : (\mathbf{x}_t, \boldsymbol{\omega}_\mathrm{o}) \rightarrow (\sigma(\mathbf{x}_t), L_\mathrm{s}(\mathbf{x}_t, \boldsymbol{\omega}_\mathrm{o}))$ with parameters $\boldsymbol{\theta}$ takes as input a position $\mathbf{x}_t$ and a direction $\boldsymbol{\omega}_\mathrm{o}$, and outputs both the density $\sigma(\mathbf{x}_t)$ at point $\mathbf{x}_t$ and the radiance $L_\mathrm{s}(\mathbf{x}_t, \boldsymbol{\omega}_\mathrm{o})$ of a light ray passing through $\mathbf{x}_t$ in direction $\boldsymbol{\omega}_\mathrm{o}$. The volume density function $\sigma(\mathbf{x}_t)$ controls the opacity at every point—large values of $\sigma(\mathbf{x}_t)$ represent opaque regions and small values represent transparent ones, which allows representation of 3D structures. The radiance function $L_\mathrm{s}(\mathbf{x}_t, \boldsymbol{\omega}_\mathrm{o})$ represents the light scattered at a point $\mathbf{x}_t$ in direction $\boldsymbol{\omega}_\mathrm{o}$, and characterizes the visual appearance of different materials (e.g., shiny or matte). Together, these two functions can be used to render images of a scene from any given camera pose. The key insight of our work is that NeRFs can be extended to model (and learn from) the raw images of a ToF camera.

NeRF optimization requires neural volume rendering: given the pose of a camera, the procedure generates an image by tracing rays through the volume and computing the radiance observed along each ray [32]:

$$L(\mathbf{x}, \boldsymbol{\omega}_\mathrm{o}) = \int_{t_\mathrm{n}}^{t_\mathrm{f}} T(\mathbf{x}, \mathbf{x}_t) \sigma(\mathbf{x}_t) L_\mathrm{s}(\mathbf{x}_t, \boldsymbol{\omega}_\mathrm{o}) \, dt, \quad \text{where} \quad T(\mathbf{x}, \mathbf{x}_t) = e^{-\int_{t_\mathrm{n}}^{t} \sigma(\mathbf{x} - \boldsymbol{\omega}_\mathrm{o} s) \, ds} \tag{1}$$

describes the transmittance for light propagating from position $\mathbf{x}$ to $\mathbf{x}_t = \mathbf{x} - \boldsymbol{\omega}_\mathrm{o} t$, for near and far bounds $t \in [t_\mathrm{n}, t_\mathrm{f}]$.

In practice, this integral is evaluated using quadrature [32]:

$$L(\mathbf{x}, \boldsymbol{\omega}_\mathrm{o}) \approx \sum_{k=1}^{N} \hat{T}(\mathbf{x}, \mathbf{x}_k)(1 - e^{-\sigma(\mathbf{x}_k)\Delta\mathbf{x}_k}) L_\mathrm{s}(\mathbf{x}_k, \boldsymbol{\omega}_\mathrm{o}), \text{ where } \hat{T}(\mathbf{x}, \mathbf{x}_k) = \prod_{j=1}^{k-1} e^{-\sigma(\mathbf{x}_j)\Delta\mathbf{x}_j}. \tag{2}$$

The value for $\Delta\mathbf{x}_j = \|\mathbf{x}_{j+1} - \mathbf{x}_j\|$ is the distance between two quadrature points.

Generalizing the neural volume rendering procedure for ToF cameras requires two changes. First, because ToF cameras use an active light source to illuminate the scene, we must consider the fact that the lighting conditions of the scene change with the position of the camera. In Section 3.1, we derive the scene's appearance in response to collocating a point light source with a camera, which follows a similar derivation to that of Bi et al. [5]. Second, in Section 3.2, we extend the volume rendering integral to model images captured with a ToF camera. Similar to the approaches taken in transient rendering frameworks [17, 39] and by neural transient fields (NeTFs) [46], we incorporate a path length importance function into our integral that can model different types of ToF cameras.

For simplicity, we assume that the function $L(\mathbf{x}, \boldsymbol{\omega}_\mathrm{o})$ is monochromatic, i.e., it outputs radiance at a single wavelength. Later on, we will model output values for red, green, blue, and infrared light (IR). $L_\mathrm{RGB}$ values correspond to radiance from ambient illumination scattering towards a color camera, whereas $L_\mathrm{ToF}$ corresponds to the measurements made by a ToF camera with active illumination.

**3.1. Collocated Point Light Source.** An ideal ToF camera responds only to the light from a collocated IR point source and not to any ambient illumination. With this assumption, we model radiance $L_\mathrm{s}(\mathbf{x}_t, \boldsymbol{\omega}_\mathrm{o})$ as a function of the source position [5]:

$$L_\mathrm{s}(\mathbf{x}_t, \boldsymbol{\omega}_\mathrm{o}) = \int_{\mathbb{S}^2} f_\mathrm{p}(\mathbf{x}_t, \boldsymbol{\omega}_\mathrm{i}, \boldsymbol{\omega}_\mathrm{o}, \mathbf{n}(\mathbf{x}_t)) L_\mathrm{i}(\mathbf{x}_t, \boldsymbol{\omega}_\mathrm{i})\, d\boldsymbol{\omega}_\mathrm{i}, \tag{3}$$

where the function $L_\mathrm{i}(\mathbf{x}_t, \boldsymbol{\omega}_\mathrm{i})$ represents the incident illumination from direction $\boldsymbol{\omega}_\mathrm{i}$, $\mathbb{S}^2$ is the unit sphere of incident directions, and the scattering phase function $f_\mathrm{p}(\mathbf{x}_t, \boldsymbol{\omega}_\mathrm{i}, \boldsymbol{\omega}_\mathrm{o}, \mathbf{n}(\mathbf{x}_t))$ describes how light is scattered at a point $\mathbf{x}_t$ in the volume. Note that the scattering phase function also depends on the local surface shading normal $\mathbf{n}(\mathbf{x}_t)$. For a point light source at $\mathbf{x}$ (i.e., collocated with the camera), each scene point is only lit from one direction. Thus, the incident radiance is given by

$$L_\mathrm{i}(\mathbf{x}_t, \boldsymbol{\omega}_\mathrm{i}) = \frac{I}{\|\mathbf{x} - \mathbf{x}_t\|^2} \delta\left(\frac{\mathbf{x} - \mathbf{x}_t}{\|\mathbf{x} - \mathbf{x}_t\|} - \boldsymbol{\omega}_\mathrm{i}\right) T(\mathbf{x}, \mathbf{x}_t), \tag{4}$$

where the scalar $I$ represents the emitted radiant intensity of the light source, $1/\|\mathbf{x}-\mathbf{x}_t\|^2$ is the inverse square light fall-off, and $\delta(\cdot)$ is the Dirac distribution used to describe only the light from a single direction. This model ignores forward scattering, which is reasonable if the scene consists mostly of completely opaque surfaces. When substituted into Equation 1 and Equation 3, the resulting forward model is

$$L(\mathbf{x}, \boldsymbol{\omega}_\mathrm{o}) = \int_{t_\mathrm{n}}^{t_\mathrm{f}} \frac{T(\mathbf{x}, \mathbf{x}_t)^2}{\|\mathbf{x} - \mathbf{x}_t\|^2} \sigma(\mathbf{x}_t) I_\mathrm{s}(\mathbf{x}_t, \boldsymbol{\omega}_\mathrm{o})\, dt \ \text{ where } \ I_\mathrm{s}(\mathbf{x}_t, \boldsymbol{\omega}_\mathrm{o}) = f_\mathrm{p}(\mathbf{x}_t, \boldsymbol{\omega}_\mathrm{o}, \boldsymbol{\omega}_\mathrm{o}, \mathbf{n}(\mathbf{x}_t)) I. \tag{5}$$

where $\boldsymbol{\omega}_\mathrm{i} = \boldsymbol{\omega}_\mathrm{o}$ in the scattering phase function $f_\mathrm{p}$ as emitted light is reflected along the same ray.

This expression is similar to Equation 1 with two key differences: the squared transmittance term, and the inverse square falloff induced by the point light source. Similar to NeRF [32], we can once again numerically approximate the above integral using quadrature, and recover the volume parameters $(\sigma(\mathbf{x}_t), I_\mathrm{s}(\mathbf{x}_t, \boldsymbol{\omega}_\mathrm{o}))$ by training a neural network that depends only on position and direction.

**3.2. Continuous-Wave ToF Model.** ToF cameras use the travel time of light to compute distances [14]. The collocated point light source sends an artificial light signal into an environment, and a ToF sensor measures the time required for light to reflect back in response. Given the constant speed of light, $\mathrm{c} \approx 3 \cdot 10^8$ m/s, this temporal information determines the distance traveled. These devices have found widespread adoption from autonomous vehicles [25] to mobile AR applications [19, 21].

Photorealistic simulations of ToF cameras involve introducing a path length importance function to the rendering equation [17, 39], and can be just as easily applied to the integral in Equation 5:

$$L_\mathrm{ToF}(\mathbf{x}, \boldsymbol{\omega}_\mathrm{o}) = \int_{t_\mathrm{n}}^{t_\mathrm{f}} \frac{T(\mathbf{x}, \mathbf{x}_t)^2}{\|\mathbf{x} - \mathbf{x}_t\|^2} \sigma(\mathbf{x}_t) I_\mathrm{s}(\mathbf{x}_t, \boldsymbol{\omega}_\mathrm{o}) W(2\|\mathbf{x} - \mathbf{x}_t\|)\, dt, \tag{6}$$

where the function $W(d)$ weights the contribution of a light path of length $d$. Note that light travels twice the distance between the camera's origin $\mathbf{x}$ and the scene point $\mathbf{x}_t$. As described by Pediredla et al. [39], the function $W(d)$ can be used to represent a wide variety of ToF cameras, including both pulsed ToF sensors [18] and continuous-wave ToF (C-ToF) sensors [12, 19, 35]. Here, as our proposed system uses a C-ToF sensor for imaging, the images are modeled using the phasor $W(d) = \exp\left(\mathrm{i}\frac{2\pi df}{\mathrm{c}}\right)$, where $f$ is the modulation frequency of the signal emitted by the C-ToF camera. Note that, because the function $W(d)$ is complex-valued, the radiance $L_\mathrm{ToF}(\mathbf{x}, \boldsymbol{\omega}_\mathrm{o})$ will also produce a complex-valued phasor image [12]. In practice, phasor images are created by capturing four real-valued images that are linearly combined (see supplemental document for additional details). In Figure 1(c), we show the real component of the phasor image, with positive pixel values as red, and negative values as blue.

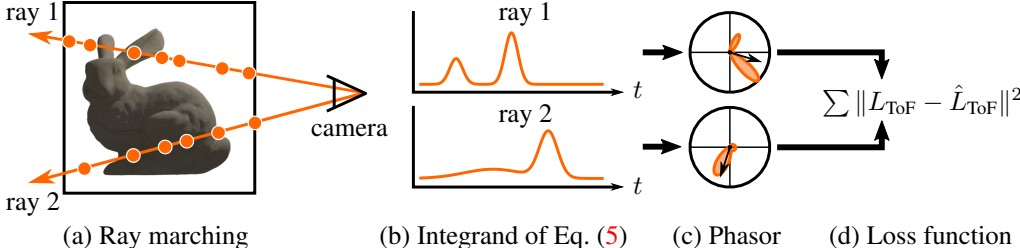

| (a) Ray marching | (b) Integrand of Eq. (5) | (c) Phasor | (d) Loss function |

Figure 2: *Neural volume rendering of C-ToF images.* **(a)** We start with ray marching to evaluate the radiance and opacity at different points along the ray. **(b)** These samples represent the continuous integrand of Equation 5, which describes the contribution of every point $\mathbf{x}_t$ along the ray. For example, ray 1 grazes both bunny ears, producing two distinct responses. **(c)** As described by Equation 6, we multiply the integrand with a complex exponential, with the result represented here in the complex plane. Integrating this result produces a single complex phasor, represented here by a vector with a magnitude corresponding to reflectance and a direction corresponding to the phase (or distance). **(d)** The loss function compares rendered phasors with the raw measurements of a C-ToF camera.

**Contrasting with ToF-derived depth.** ToF cameras typically recover depth by assuming only one point $\mathbf{x}_s$ reflects light for every ray, i.e., the integrand of Equation 6 is assumed to be zero for all other points $\mathbf{x} \neq \mathbf{x}_s$ (points in front of $\mathbf{x}_s$ reflect no light, and points behind $\mathbf{x}_s$ are hidden). Under these assumptions, Equation 6 simplifies to the phasor:

$$L_{\text{ToF}}(\mathbf{x}, \boldsymbol{\omega}_o) = a \cdot W(2\|\mathbf{x} - \mathbf{x}_s\|) = a \cdot \exp\left(\mathrm{i}\frac{4\pi f}{c}\|\mathbf{x} - \mathbf{x}_s\|\right), \tag{7}$$

where the phasor's magnitude, $|L_{\text{ToF}}(\mathbf{x}, \boldsymbol{\omega}_o)| = a$, represents the amount of light reflected by this single point, and the phase, $\angle L_{\text{ToF}}(\mathbf{x}, \boldsymbol{\omega}_o) = \frac{4\pi f}{c}\|\mathbf{x} - \mathbf{x}_s\| \bmod 2\pi$, is related to distance $\|\mathbf{x} - \mathbf{x}_s\|$.

In real-world scenarios, it is also possible for multiple points along a ray to contribute to the signal, resulting in a linear combination of phasor radiance values—known as multi-path interference. This can degrade the quality of depth measurements for a C-ToF camera. For example, around depth edges, a pixel integrates the signal from surfaces at two different distances from the camera (e.g., Figure 2), resulting in 'flying pixel' artifacts [43] (i.e., 3D points not corresponding to either distance). Similar artifacts occur when imaging semi-transparent or specular objects, where two or more surfaces contribute light to a pixel.

Optimizing NeRFs with phasor images via Equation 6 therefore has three distinct advantages over using derived depth maps via Equation 7. (i) For ranges that span values larger than $\frac{c}{2f}$, the true range is ambiguous, as there are multiple depth values that produce the same phase. For example, a typical modulation frequency of $f = 30\,\text{MHz}$ for a C-ToF camera corresponds to an unambigous range of $\frac{c}{2f} \approx 5\,\text{m}$. By modeling the phasor images directly, we avoid the issues associated with recovering depth images for scenes that exceed this range (Figure 3). (ii) Depth values become unreliable (noisy) when the amount of light reflected to the camera is small. Modeling the phasor images directly makes the solution robust to sensor noise (Figure 4). (iii) For regions near depth edges (Figure 2) or for objects with complicated reflectance properties (e.g., transparent or specular surfaces), the light detected may not travel along a single path; this results in mixtures of phasors, producing phase values that do not correspond to a single depth. Equation 6 models the response from multiple single-scattering events along a ray, providing us with a better handle over such scenarios.

## 4 Optimizing Dynamic ToF + NeRF = TöRF

**4.1. Dynamic Neural Radiance Fields.** One key advantage of working with phasor images is that the method can capture scene geometry from a single view, which enables higher-fidelity novel-view synthesis of dynamic scenes from a potentially moving color camera and C-ToF camera pair. To support dynamic neural radiance fields, we model the measurements with two neural networks. The first, static network $F_\theta^{\text{stat}} : (\mathbf{x}_t, \boldsymbol{\omega}_o) \rightarrow (\sigma^{\text{stat}}(\mathbf{x}_t), L_s^{\text{stat}}(\mathbf{x}_t, \boldsymbol{\omega}_o), I_s^{\text{stat}}(\mathbf{x}_t, \boldsymbol{\omega}_o))$ is a 5D function of position and direction, while the second, dynamic network $F_\theta^{\text{dyn}} : (\mathbf{x}_t, \boldsymbol{\omega}_o, \tau) \rightarrow (\sigma^{\text{dyn}}(\mathbf{x}_t, \tau), L_s^{\text{dyn}}(\mathbf{x}_t, \boldsymbol{\omega}_o), I_s^{\text{dyn}}(\mathbf{x}_t, \boldsymbol{\omega}_o, \tau), b(\mathbf{x}_t, \tau))$ is a 6D function of position, direction, and time $\tau$. Instead of directly consuming a time $\tau$, the dynamic network receives a latent code $\mathbf{z}_\tau$ which is

| Color (input) | Depth (input) | Color (TöRF) | Depth (TöRF) |

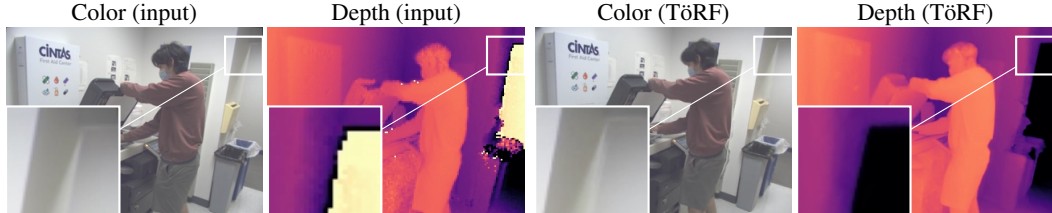

Figure 3: *Raw phasor supervision avoids wrap-around errors.* Wrap-around phase bounds the range of useful ToF measurements (left), causing errors when depth is used as supervision. Our approach of modeling raw phasor measurements within the neural volume alleviates this problem (right). This is because only one phase offset is consistent across multiple camera views.

| Color (input) | Depth (input) | Color (TöRF) | Depth (TöRF) |

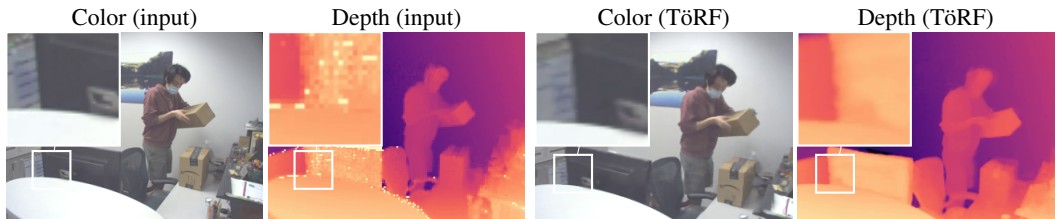

Figure 4: *Raw phasor supervision reduces noise in dark objects.* The weak signal reflected back by dark objects (e.g., the computer monitor) results in noisy depth measurements. As TöRF does not rely on depth explicitly and instead models raw phasor images, our recovered depth map better captures the scene geometry in comparison to ToF-derived depth. This is because when phasor magnitudes are small, TöRF falls back on triangulation cues to recover geometry.

optimized per frame, similar to Li et al. [24]. Following the approach of Li et al. [26], we blend the outputs of the static and dynamic networks using a position- and time-dependent blending weight $b(\mathbf{x}_t, \tau)$ that is predicted by the dynamic network $F_{\boldsymbol{\theta}}^{\mathrm{dyn}}$, as in Gao et al. [10]. This produces density $\sigma^{\mathrm{blend}}$, radiance $L_{\mathrm{s}}^{\mathrm{blend}}$, and radiant intensity $I_{\mathrm{s}}^{\mathrm{blend}}$ values to pass into our image formation models:

$$L_{\mathrm{RGB}}(\mathbf{x}, \boldsymbol{\omega}_{\mathrm{o}}, \tau) = \int_{t_{\mathrm{n}}}^{t_{\mathrm{f}}} T^{\mathrm{blend}}(\mathbf{x}, \mathbf{x}_t, \tau) \sigma^{\mathrm{blend}}(\mathbf{x}_t, \tau) L_{\mathrm{s}}^{\mathrm{blend}}(\mathbf{x}_t, \boldsymbol{\omega}_{\mathrm{o}}, \tau) \, dt \tag{8}$$

$$L_{\mathrm{ToF}}(\mathbf{x}, \boldsymbol{\omega}_{\mathrm{o}}, \tau) = \int_{t_{\mathrm{n}}}^{t_{\mathrm{f}}} \frac{T^{\mathrm{blend}}(\mathbf{x}, \mathbf{x}_t, \tau)^2}{\|\mathbf{x} - \mathbf{x}_t\|^2} \sigma^{\mathrm{blend}}(\mathbf{x}_t, \tau) I_{\mathrm{s}}^{\mathrm{blend}}(\mathbf{x}_t, \boldsymbol{\omega}_{\mathrm{o}}, \tau) W(2 \|\mathbf{x} - \mathbf{x}_t\|) \, dt. \tag{9}$$

See the supplemental document for an explicit definition of the blending terms.

**4.2. Loss Function.** Given a set of color images and phasor images captured of a scene at different time instances, we sample a set of camera rays from the set of all pixels, and minimize the following total squared error between the rendered images and measured pixel values:

$$\mathcal{L} = \sum_{(\mathbf{x}, \boldsymbol{\omega}_{\mathrm{o}}, \tau)} \|L_{\mathrm{RGB}}(\mathbf{x}, \boldsymbol{\omega}_{\mathrm{o}}, \tau) - \hat{L}_{\mathrm{RGB}}(\mathbf{x}, \boldsymbol{\omega}_{\mathrm{o}}, \tau)\|^2 + \lambda \|L_{\mathrm{ToF}}(\mathbf{x}, \boldsymbol{\omega}_{\mathrm{o}}, \tau) - \hat{L}_{\mathrm{ToF}}(\mathbf{x}, \boldsymbol{\omega}_{\mathrm{o}}, \tau)\|^2, \tag{10}$$

where the scalar $\lambda \geq 0$ controls the relative contribution of both loss terms, $\hat{L}_{\mathrm{RGB}}(\mathbf{x}, \boldsymbol{\omega}_{\mathrm{o}}, \tau)$ represents the measurements of a color camera, and $\hat{L}_{\mathrm{ToF}}(\mathbf{x}, \boldsymbol{\omega}_{\mathrm{o}}, \tau)$ represents the phasor measurements of a C-ToF camera. At training time, we reduce the weight $\lambda$ in later iterations to prioritize the color loss (halved every 125,000 iterations).

**4.3. Camera Pose Optimization.** In past works, COLMAP [45] has been used to recover camera poses for NVS. However, COLMAP fails to recover accurate camera poses for many real scenes even if we masked dynamic regions [20]. Further, COLMAP only recovers camera poses up to unknown scale, whereas our ToF image formation model assumes a known scene scale. As such, for real-world scenes, we optimize camera poses from scratch within the training loop. First, we optimize the weights of the static neural network $F_{\boldsymbol{\theta}}^{\mathrm{stat}}$, as well as the camera poses for each video frame and the relative rotation and translation between the color and C-ToF sensor, with a learning rate of $10^{-3}$. After 5000 iterations, we decrease the pose learning rate to $5 \cdot 10^{-4}$, and optimize our full model.

**4.4. Ray Sampling.** Many physical camera systems do not have collocated color and ToF cameras. As such, to train our model, we trace separate rays through the volume for color and ToF measurements. We alternate using the color loss and the ToF loss for every iteration. Further, like NeRF [32], we use stratified random sampling when sampling points along a ray.

## 5 Experiments

**5.1. Hardware.** We use an iDS UI-3070CP-C-HQ machine vision camera to provide RGB measurements (640×480 @ 30 fps; downsampled to 320×240), and a Texas Instruments OPT8241 sensor to provide phasor measurements (320×240 @ 30 fps) with an unambiguous range of 5 m. Both cameras are mounted with a baseline of 41 mm. We use OpenCV to calibrate intrinsics, extrinsics, and lens distortion. See the supplement for details.

For optimization, we use an NVIDIA GeForce RTX 2080 Ti with 11 GB RAM. Our model takes 12–24 hours to converge, and 3–5 seconds per frame to generate a novel view (256×256).

**5.2. Data.** We captured the *PhoneBooth*, *Cupboard*, *Photocopier*, *DeskBox*, and *StudyBook* sequences with our handheld camera setup. Each is indoors in an office with a person performing a dynamic action, and includes view dependence from real-world materials. *PhoneBooth* includes multi-path interference effects from a glass door, and *Photocopier* includes wrap-around phase effects in the distance. For comparison, we also captured the *Dishwasher* sequence on an iPhone 12 Pro, which uses a ToF sensor to capture depth (raw measurements are not available). Finally, we create synthetic raw C-ToF sequences *Bathroom*, *Bedroom*, and *DinoPear* by adapting the physically-based path tracer PBRT [40] to generate phasor images with multi-bounce and scattering effects.

**5.3. Few-View Reconstruction of Static Scenes.** We demonstrate that integrating raw ToF measurements in addition to RGB enables TöRF to reconstruct static scenes from fewer input views, and to achieve higher visual fidelity than standard NeRF [32] for the same number of input views. Table 2 contains a quantitative comparison on two synthetic sequences, *Bathroom* and *Bedroom*, for reconstructions from just 2 and 4 input views. To enable the comparison on 10 hold-out views, we use ground-truth camera poses for both methods. With just two input views, TöRF's added phasor supervision better reproduces the scene than NeRF, as one might expect. This closely resembles a camera system that might exist on a smartphone, and shows the potential value of ToF supervision for dynamic scenes if we consider a static scene as one time step of a video sequence. For four views, NeRF and TöRF produce comparable RGB results, though our depth reconstructions are significantly more accurate (Figure 5).

**5.4. Dynamic Scenes.** We compare reconstruction quality on the synthetic dynamic sequence *DinoPear* in Table 3 with 30 ground-truth hold-out views and depth maps. Compared to methods that use deep depth estimates (NSFF and VideoNeRF), TöRF produces better depth and RGB views. While TöRF PSNR is slightly lower than NSFF's, the perceptual LPIPS metric is significantly lower for TöRF, which matches the findings from our qualitative results. TöRF also produces better depth and RGB reconstructions than the same methods modified to use ToF-derived depth (NSFF+ToF, VideoNeRF+ToF).

For real-world scenes, we show results and comparisons in Figure 6. VideoNeRF+ToF shows stronger disocclusion artifacts and warped edges near depth boundaries, and cannot recover from depth maps with wrapped range. NSFF suffers from severe ghosting and stretching artifacts that negatively impact the quality of the results. Our results show the highest visual quality and most accurate depth maps. Please see the videos on our website for in-motion novel-view synthesis.

## 6 Discussion

**6.1. Limitations.** Introducing ToF sensors into RGB neural radiance fields aims to improve quality by merging the benefits of both sensing modes; but, some limitations are also brought in through ToF sensing. C-ToF sensing can struggle on larger-scale scenes; however, using multiple different modulation frequencies can extend the unambiguous range [12]. Using different coding methods can also increase depth precision [13]. While C-ToF sensors typically struggle outdoors, EpiToF [1] has demonstrated the ability to perform 15 m ranging under strong ambient illumination. Further, for each measurement, C-ToF sensors require capturing four or more images quickly at different times, which can cause artifacts for fast-moving objects.

Table 2: *Phasor supervision aids few-view reconstruction.* Each cell contains RGB image similarity measures, and metrics are computed on 10 hold-out views. TöRF significantly outperforms NeRF on both synthetic static scenes and produces more accurate depth estimates (Figure 5), particularly from just two input views. Note that the metric depth error 'MSE (D)' is affected by mirrors, particularly in the bathroom scene, whose depth is defined by the plane of the mirror and not the objects in its reflection.

| Views | Method | *Bathroom* | | | | *Bedroom* | | | |
| --- | --- | --- | --- | --- | --- | --- | --- | --- | --- |
| | | MSE (D) ▼ | PSNR ▲ | SSIM ▲ | LPIPS ▼ | MSE (D) ▼ | PSNR ▲ | SSIM ▲ | LPIPS ▼ |
| 2 | NeRF [32] | **0.97** | 16.56 | 0.660 | 0.022 | 18.43 | 11.59 | 0.313 | 0.056 |
| | TöRF (ours) | 2.12 | **19.21** | **0.739** | **0.015** | **0.31** | **22.09** | **0.840** | **0.010** |
| 4 | NeRF [32] | **0.70** | 24.17 | 0.864 | **0.008** | 0.94 | 28.29 | 0.936 | 0.003 |
| | TöRF (ours) | 0.76 | **26.18** | **0.879** | 0.009 | **0.27** | **29.79** | **0.938** | **0.002** |

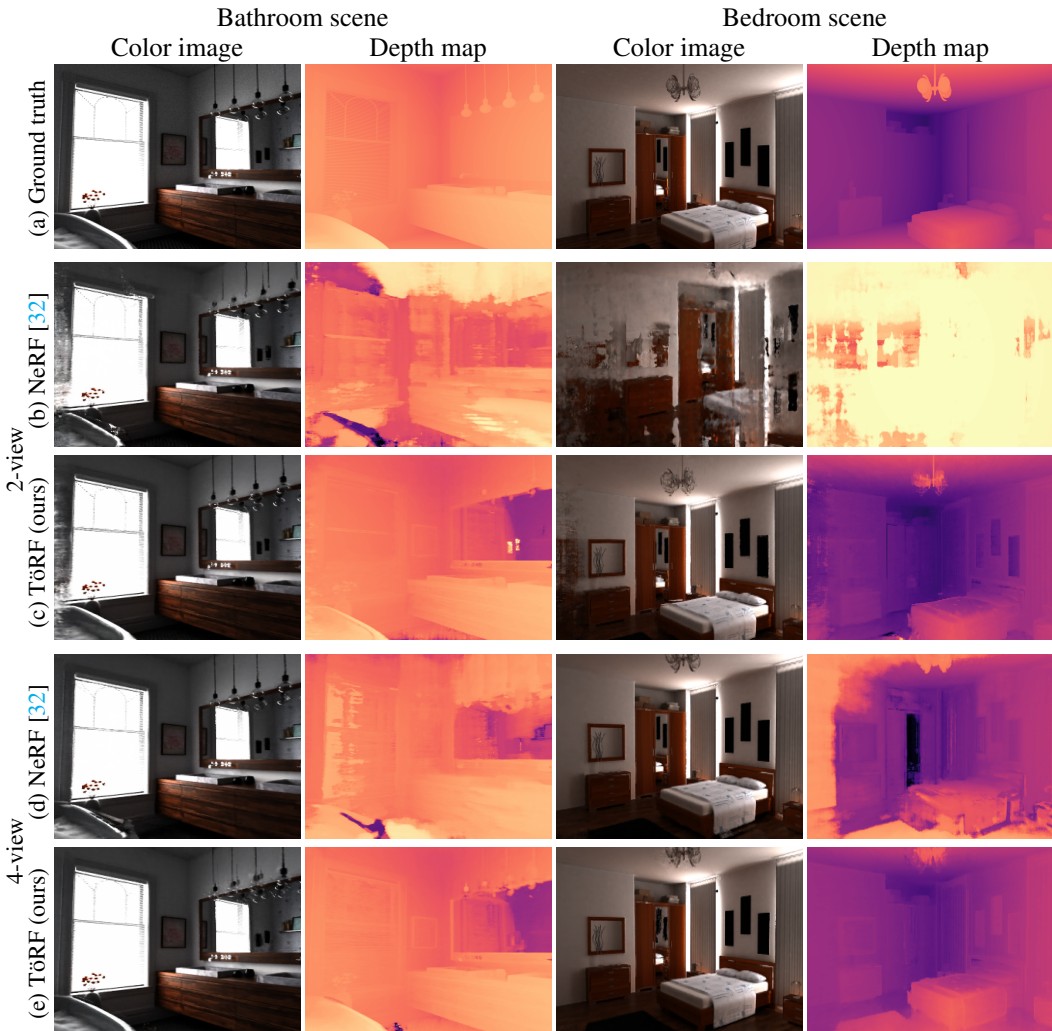

Figure 5: *Adding ToF aids reconstruction for few views in static scenes.* NeRF quality suffers as the number of views decreases, but adding ToF data makes view synthesis possible with two RGB views. Note the cleaner depth, sharper edges, and thin geometric details such as the lamps above the mirror.

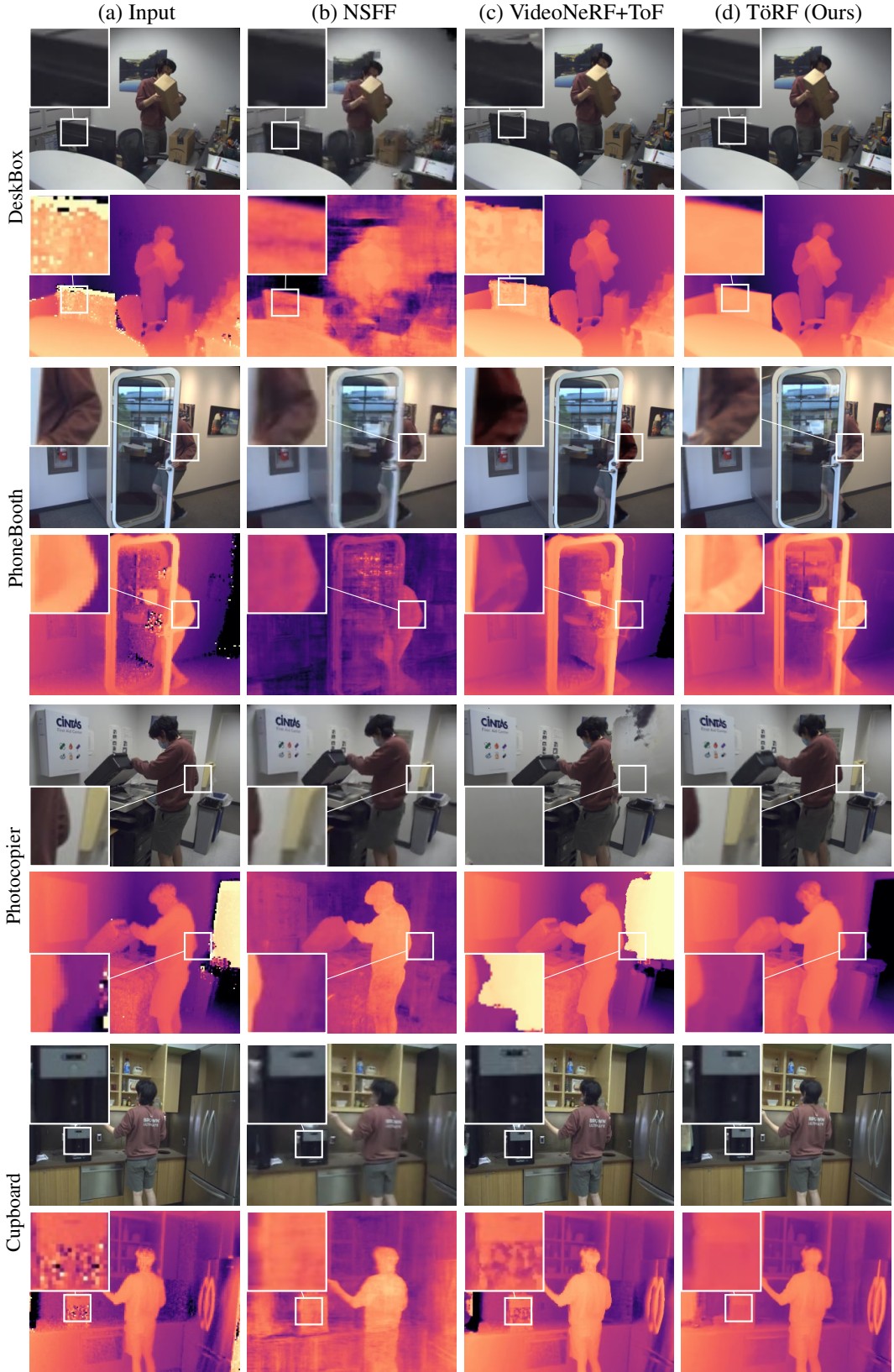

Figure 6: *Adding ToF supervision helps improve quality.* Compared to two video baselines, Video-NeRF [52] (modified to use ToF-derived depth) and NSFF [26], our approach reduces errors in static scene components and some dynamic components. All models were trained for comparable times. Please see the videos on our website for additional comparisons in motion.

Table 3: Evaluation on ground-truth hold-out views for the dynamic *DinoPear* sequence shows improved depth and RGB results for our method. Note that NSFF and VideoNeRF are given manually unwrapped ToF depth produced by adding $2\pi$ to all phase values below a certain threshold. The TöRF approach of using raw phasor images produces better depth reconstructions. While NSFF produces the highest PSNR, this does not match the perceived visual quality—please see our webpage.

| Method | Depth MSE ▼ | PSNR ▲ | SSIM ▲ | LPIPS ▼ |
|---|---|---|---|---|
| NSFF [26] | $0.021 \pm 0.003$ | $\mathbf{22.64 \pm 1.46}$ | $0.554 \pm 0.029$ | $0.039 \pm 0.010$ |
| + ToF depth | $0.010 \pm 0.002$ | $21.84 \pm 0.72$ | $0.382 \pm 0.021$ | $0.037 \pm 0.014$ |
| + ToF depth (unwrapped) | $0.007 \pm 0.002$ | $21.70 \pm 0.98$ | $0.387 \pm 0.028$ | $0.040 \pm 0.013$ |
| VideoNeRF [52] | $0.008 \pm 0.002$ | $21.32 \pm 1.03$ | $0.358 \pm 0.032$ | $0.032 \pm 0.017$ |
| + ToF depth | $0.011 \pm 0.002$ | $19.75 \pm 1.07$ | $0.275 \pm 0.021$ | $0.041 \pm 0.016$ |
| + ToF depth (unwrapped) | $0.009 \pm 0.002$ | $20.72 \pm 1.03$ | $0.350 \pm 0.033$ | $0.032 \pm 0.016$ |
| TöRF (ours) | $\mathbf{0.005 \pm 0.001}$ | $22.19 \pm 1.75$ | $\mathbf{0.561 \pm 0.052}$ | $\mathbf{0.028 \pm 0.011}$ |

Even with ToF data, objects imaged at grazing angles or objects that are both dark (low reflectance) and dynamic remain difficult to reconstruct, e.g., dark hair (Figure 7). Further, neural networks have limited capacity to model dynamic scenes, which limits the duration of dynamic sequences. This is a limitation of many current neural dynamic scene methods.

**6.2. Potential Social Impact.** Scene reconstruction and view synthesis are core problems in visual computing for determining the shape and appearance of objects and scenes. Neural approaches to these tasks hold promise to increase accuracy and fidelity. At the methodological level, integrating ToF data improves accuracy, but restricts use to scenarios where active illumination is detectable. While the recovery of shape and appearance has many applications, negative impact may include synthesizing images from perspectives or time instances that were never captured (falsifying media), extending surveillance through higher-fidelity reconstructions (security), or copying physical objects to 'rip off' designs.

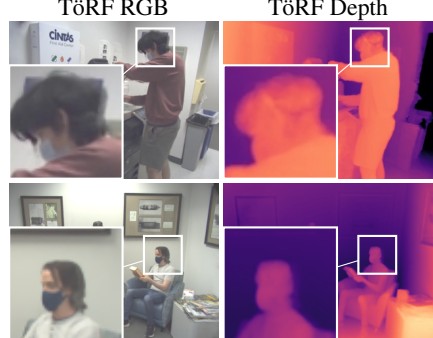

TöRF RGB    TöRF Depth

Figure 7: In the *Photocopier* (top) and *Study-Book* (bottom) scenes, RGB and depth NVS show that dark and dynamic objects pose difficulties. Dark hair incorrectly extends in front or to the side of the face due to failure to reconstruct dynamic motion.

Practically, current neural approaches are more computationally expensive in both optimization and rendering than classic image-based rendering. Our work required GPUs to optimize for many hours (12–24 h). Without renewable energy sources, this use will generate $CO_2$ emissions, requiring 1.5–3 kg $CO_2$-equivalents per scene for optimization and 0.01–0.02 kg $CO_2$-equivalents per sequence for rendering (numbers generated by ML $CO_2$ Impact [22]). Concurrent work in neural radiance fields reduces this cost using caching, spatial acceleration structures, and more efficient parameterizations, and real-world deployment should exploit these approaches to reduce the $CO_2$ emission impact.

# 7  Conclusion

Modern camera systems integrate multiple modes of sensing, and our reconstruction methods should exploit this information to improve quality. To this end, we formulate a neural model for time-of-flight radiance fields based on physical RGB+ToF image formation. We demonstrate an optimization method to recover TöRF volumes, and show that it improves novel-view synthesis for few-view scenes and especially for dynamic scenes. Further, we demonstrate that using raw ToF phasor supervision leads to better performance than using derived depth directly, allowing both sensing modes to help resolve errors, limitations, and ambiguities. Future work may extend the combination of additional sensors into neural radiance fields, e.g., dynamic vision sensors [27] or *event cameras* may be used to measure scenes at higher speeds. Further, a collocated point light source has been shown to be able to render photos of scenes under non-collocated illumination conditions [5]. As a result, we believe ToF images may also serve to support relighting applications.

## Acknowledgments and Disclosure of Funding

Thank you to Kenan Deng for developing acquisition software for the time-of-flight camera. For our synthetic data, we thank the authors of assets from the McGuire Computer Graphics Archive [29], Benedikt Bitterli for Mitsuba scene files [6], Davide Tirindelli for Blender scene files, 'Architectural Visualization' demo Blender scene by Marek Moravec (CC-0 Public Domain) [33], 'Rampaging T-Rex' from the 3D library of Microsoft's 3D Viewer, and 'Indoor Pot Plant 2' by 3dhaupt from Free3D (non-commercial) [15].

For funding, Matthew O'Toole acknowledges support from NSF IIS-2008464, James Tompkin thanks an Amazon Research Award and NSF CNS-2038897, and Christian Richardt acknowledges funding from an EPSRC-UKRI Innovation Fellowship (EP/S001050/1) and RCUK grant CAMERA (EP/M023281/1, EP/T022523/1).

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
