# TöRF: Time-of-Flight Radiance Fields
# for Dynamic Scene View Synthesis
# Supplemental Document

**Benjamin Attal**
Carnegie Mellon University

**Eliot Laidlaw**
Brown University

**Aaron Gokaslan**
Cornell University

**Changil Kim**
Facebook

**Christian Richardt**
University of Bath

**James Tompkin**
Brown University

**Matthew O'Toole**
Carnegie Mellon University

imaging.cs.cmu.edu/torf

## 1 Additional Results

More results are shown in Figure 8 for the *StudyBook* and *Dishwasher* scenes, and Figure 10 for the *DinoPear* scene. Figure 9 also highlights our ability to account for multi-path interference. We also show animated results and comparisons for all sequences on our website.

**1.1. iPhone ToF—*Dishwasher* Dynamic Scene.**    To evaluate a more practical camera setup than our prototype, we captured one real-world sequence (the *Dishwasher* scene) with a standard handheld Apple iPhone 12 Pro. This consumer smartphone contains a LIDAR ToF sensor for measuring sparse metric depth, which is processed by ARKit to provide a dense metric depth map video in addition to a captured RGB color video. Unfortunately, the raw measurements are not available from the ARKit SDK; however, if available, in principle our approach could apply.

Thus, for processing with TöRF, we convert the estimated metric depth maps to synthetic C-ToF sequences by assuming a constant infrared albedo everywhere. In this specific case, the RGB and ToF data are also collocated, as the depth maps are aligned with the color video.

## 2 Dynamic Field Blending

Here, we explain how we model dynamic scenes using the RGB case; the ToF case is similar and uses collocated reflected radiant intensity $I_\text{s}$ instead of scattered radiance $L_\text{s}$. We evaluate the integral in Equation 8 using quadrature [3] as follows:

$$L_\text{RGB}(\mathbf{x}, \boldsymbol{\omega}_\text{o}, \tau) = \sum_{k=0}^{N} \hat{T}^\text{blend}(\mathbf{x}, \mathbf{x}_k, \tau) \alpha^\text{blend}(\mathbf{x}_k, \tau) L_\text{s}^\text{blend}(\mathbf{x}_k, \boldsymbol{\omega}_\text{o}, \tau). \tag{11}$$

Here, $\hat{T}^\text{blend}$ is the blended transmittance for light propagating from $\mathbf{x}$ to $\mathbf{x}_k = \mathbf{x} - \boldsymbol{\omega}_\text{o}k$ at time $\tau$:

$$\hat{T}^\text{blend}(\mathbf{x}, \mathbf{x}_k, \tau) = \prod_{j=0}^{k-1} \left(1 - \alpha^\text{blend}(\mathbf{x}_j, \tau)\right), \tag{12}$$

where $\alpha^\text{blend}$ is the blended opacity at position $\mathbf{x}_k$ and time $\tau$. This blend combines the opacities

$$\alpha^\text{stat}(\mathbf{x}_k) = 1 - \exp(-\sigma^\text{stat}(\mathbf{x}_k)\Delta\mathbf{x}_k) \tag{13}$$

$$\alpha^\text{dyn}(\mathbf{x}_k, \tau) = 1 - \exp(-\sigma^\text{dyn}(\mathbf{x}_k, \tau)\Delta\mathbf{x}_k) \tag{14}$$

predicted by the static and dynamic networks, respectively, using the blending weight $b(\mathbf{x}_k, \tau)$:

$$\alpha^\text{blend}(\mathbf{x}_k, \tau) = (1 - b(\mathbf{x}_k, \tau)) \cdot \alpha^\text{stat}(\mathbf{x}_k) + b(\mathbf{x}_k, \tau) \cdot \alpha^\text{dyn}(\mathbf{x}_k, \tau). \tag{15}$$

35th Conference on Neural Information Processing Systems (NeurIPS 2021).

The blended radiance $L_\text{s}^\text{blend}$, premultiplied by the blended opacity $\alpha^\text{blend}$, is calculated using

$$\alpha^\text{blend}(\mathbf{x}_k, \tau) L_\text{s}^\text{blend}(\mathbf{x}_k, \boldsymbol{\omega}_\text{o}, \tau) = (1 - b(\mathbf{x}_k, \tau)) \cdot \alpha^\text{stat}(\mathbf{x}_k) L_\text{s}^\text{stat}(\mathbf{x}_k, \boldsymbol{\omega}_\text{o}) \tag{16}$$

$$+ b(\mathbf{x}_k, \tau) \cdot \alpha^\text{dyn}(\mathbf{x}_k, \tau) L_\text{s}^\text{dyn}(\mathbf{x}_k, \boldsymbol{\omega}_\text{o}, \tau), \tag{17}$$

where $L_\text{s}^\text{stat}$ and $L_\text{s}^\text{dyn}$ are the scattered radiance predicted by the static and dynamic networks. We similarly compute the radiant intensity $I_\text{s}^\text{blend}$ used by $L_\text{ToF}$ in Equation 9.

## 3 Continuous-wave Time-of-Flight Image Formation Model

A continuous-wave time-of-flight (C-ToF) sensor is an active imaging system that illuminates the scene with a point light source. The intensity of this light source is modulated with a temporally-varying function $f(t)$, and the temporally-varying response at a camera pixel is

$$i(t) = \int_{-\infty}^{\infty} R(t - s) f(s) \, ds, \tag{18}$$

where $R(t)$ is the scene's temporal response function observed at a particular camera pixel (i.e., the response to a pulse of light emitted at $t = 0$). Note that Equation 18 is a convolution operation between the scene's temporal response function $R(t)$ and the light source modulation function $f(t)$.

The operating principle of a C-ToF sensor is to modulate the exposure incident on the sensor with a function $g(t)$, and integrating the response over the exposure period. Suppose that $f(t)$ and $g(t)$ are periodic functions with period $T$, and there are $N$ periods during an exposure. A C-ToF sensor would then measure the following:

$$L = \int_0^{NT} g(t) i(t) \, dt \tag{19}$$

$$= \int_0^{NT} g(t) \left( \int_{-\infty}^{\infty} R(t - s) f(s) ds \right) dt \tag{20}$$

$$= N \int_{-\infty}^{\infty} R(s) \underbrace{\left( \int_0^T f(t - s) g(t) dt \right)}_{=h(s)} ds, \tag{21}$$

where the function $h(t)$ is the convolution between the exposure modulation function $g(t)$ and the light source modulation function $f(t)$. This function $h(t)$ can be interpreted as a path length importance function, which weights the contribution of a light path based on its path length.

In this work, we assume that the C-ToF camera produces phasor images [1], where $h(t) = \exp(\text{i} 2\pi\omega t)$. To achieve this, suppose that $f(t) = \frac{1}{2}\sin(2\pi\omega t) + \frac{1}{2}$ and $g(t) = \sin(2\pi\omega t + \phi)$ for a modulation frequency $\omega = \frac{1}{T}$, where $\phi$ is a controllable phase offset between the two signals. The convolution between these two functions is then $h(t) = \frac{T}{4}\cos(2\pi\omega t + \phi)$. After capturing four images $L_\phi$ with different phase offsets $\phi \in \{0, \frac{\pi}{2}, \pi, \frac{3\pi}{2}\}$, we can linearly recombine these measurements as follows:

$$L_\text{ToF} = (L_0 - L_\pi) - \text{i}(L_{\frac{\pi}{2}} - L_{\frac{3\pi}{2}}) = \frac{NT}{2} \int_{-\infty}^{\infty} R(s) \exp(\text{i} 2\pi\omega s) \, ds. \tag{22}$$

The response at every pixel is therefore a complex phasor. Figure 11(b) and Figure 11(c) provide an example of the real and imaginary component of this phasor image, respectively. As discussed in the main paper, in typical depth sensing scenarios, the phasor's magnitude, $|L_\text{ToF}|$, represents the amount of light reflected by a single point in the scene (Figure 11(e)), and the phase, $\angle L_\text{ToF}$, is related to distance of that point (Figure 11(f)).

## 4 Experimental C-ToF Setup

The hardware setup shown in Figure 11(a) consists of a standard machine vision camera and a time-of-flight camera. Our USB 3.0 industrial color camera (UI-3070CP-C-HQ Rev. 2) from iDS has a sensor resolution of $2056 \times 1542$ pixels, operates at 30 frames per second, and uses a 6 mm lens with an $f/1.2$ aperture. Our high-performance time-of-flight camera (OPT8241-CDK-EVM)

from Texas Instruments has a sensor resolution of 320×240 pixels, and also operates at 30 frames per second (software synchronized with the color camera). Camera exposure was 10 ms. The illumination source wavelength of the time-of-flight camera is infrared (850 nm) and invisible to the color camera. The modulation frequency of the time-of-flight camera is $\omega = 30$ MHz, resulting in an unambiguous range of 5 m. Both cameras are mounted onto an optical plate, and have a baseline of approximately 41 mm.

We use OpenCV to calibrate the intrinsics, extrinsics and distortion coefficients of the stereo camera system. We undistort all captured images, and resize the color image to 640×480 to improve optimization performance. In addition, the phase associated with the C-ToF measurements may be offset by an unknown constant; we recover this common zero-phase offset by comparing the measured phase values to the recovered position of the calibration target. For simplicity, we assume that the modulation frequency associated with the C-ToF camera is an approximately sinusoidal signal, and ignore any nonlinearities between the recovered phase measurements and the true depth.

Along with the downsampled 640×480 color images, the C-ToF measurements consist of the four 320×240 images, each representing the scene response to a different predefined phase offset $\phi$. We linearly combine the four images into a complex-valued C-ToF phasor image representing the response to a complex light signal, as described in Equation 22. To visualize these complex-valued phasor images, we show the real component and imaginary component separately, and label positive pixel values as red and negative values as blue.

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

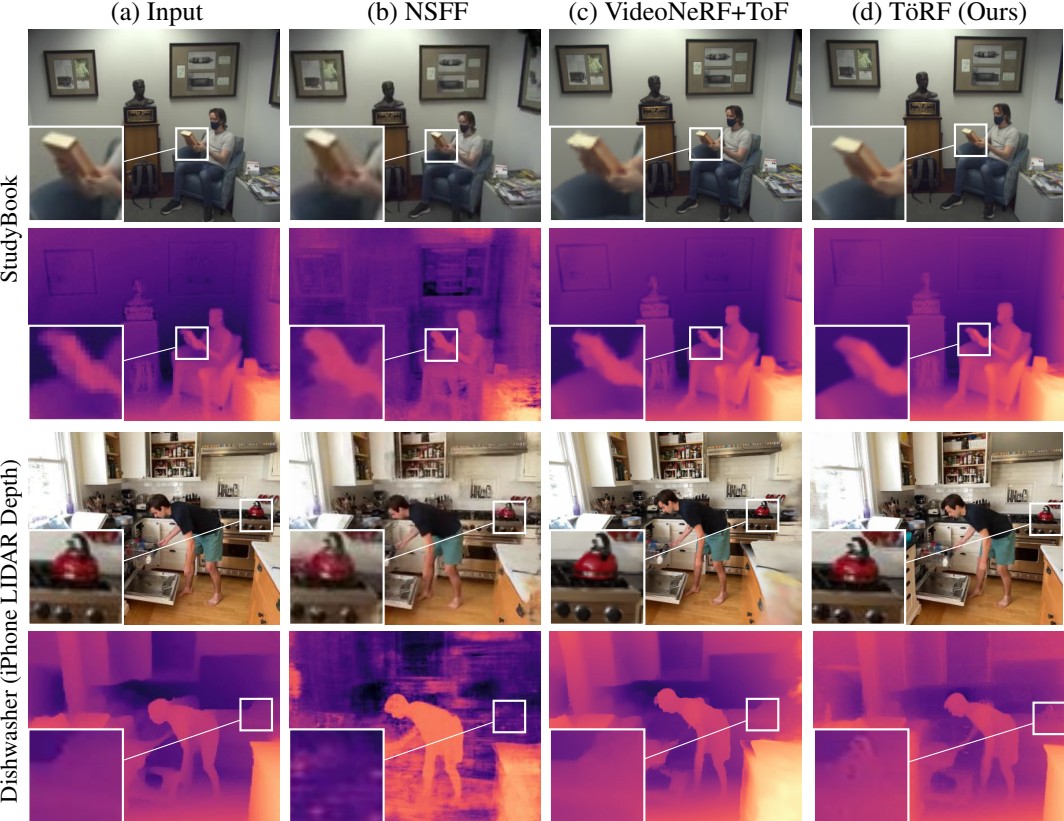

Figure 8: *Adding ToF supervision helps improve quality.* Compared to two video baselines, Video-NeRF [4] (modified to use ToF-derived depth) and NSFF [2], our approach reduces errors in static scene components and some dynamic components. All models were trained for comparable times. Please see our website for video comparisons.

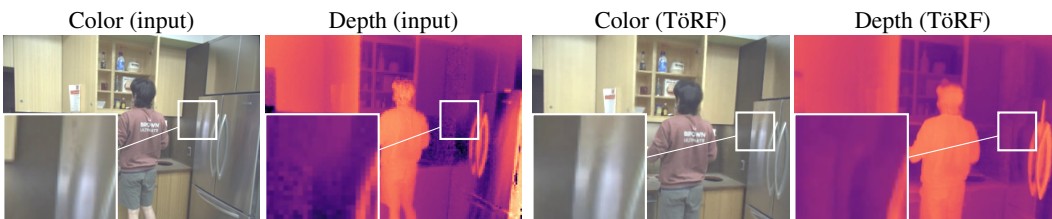

Figure 9: *Raw phasor supervision accounts for multi-path interference.* Specular reflections off of the metallic fridge door result in a C-ToF depth image that does not accurately capture the true geometry of the door. Instead, the C-ToF camera captures a mixture of phasors: phasors representing the surface of the fridge door, and phasors representing the virtual (reflected) image of objects seen in the fridge door. The resulting mixture of phasors biases the depth values such that the fridge door appears to be further away from the camera. It is important to note that the virtual image of objects moves as though the objects were being imaged directly (e.g., as in the case of a mirror). TöRF accounts for multi-path interference by modeling the raw phasor image as a summation of different phasors, and more effectively predicts the appearance of such complex scenes for novel viewpoints.

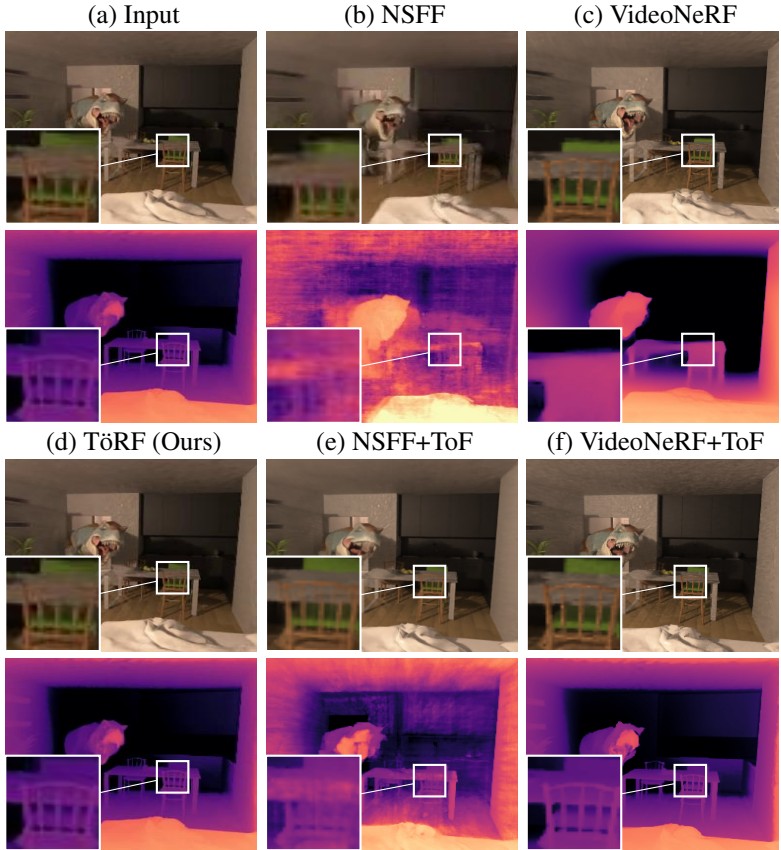

Figure 10: *Comparisons to baseline algorithms.* We compare TöRF results to those from NSFF [2], VideoNeRF [4], and modified versions of both algorithms that take unwrapped depth images as input. All models were trained for comparable times. Please see our website for video comparisons.

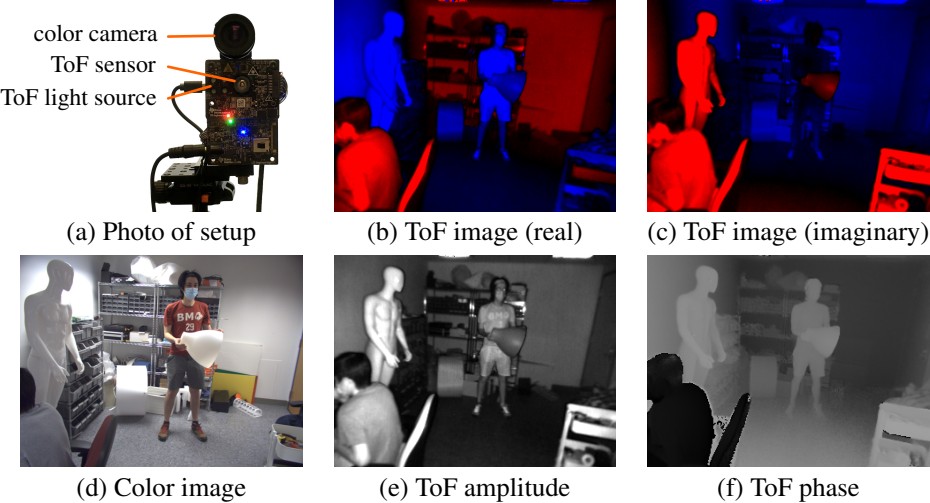

Figure 11: **(a)** Photo of the proposed hardware setup, consisting of a single ToF and a color camera. **(b)** Real component of ToF phasor image (positive/negative values), captured with a modulation frequency $\omega = 30$ MHz. **(c)** Imaginary component of ToF phasor image. **(d)** Color image from color camera. **(e)** Amplitude of the phasor image; represents the average amount of infrared light reflected by the scene. **(f)** Phase of the phasor image; values are approximately proportional to range.