# OpenReview forum: "TöRF: Time-of-Flight Radiance Fields for Dynamic Scene View Synthesis"
_NeurIPS.cc/2021/Conference — NeurIPS 2021 Poster_

### Official Review · Reviewer_EBPE · 2021-06-25

**Rating:** 6
**Confidence:** 4

**Summary:**

The goal is geometry and appearance reconstruction of a dynamic scene from monocular RGB and C-ToF (raw depth) measurements. The proposed method extends dynamic NeRF models by a principled image formation model of C-ToF measurements. It is assumed that only the ToF camera light enters the ToF camera sensor, that both are at the same position, and that there is only one bounce per light path. Experiments on few-view static reconstruction and monocular dynamic reconstruction are included.

**Limitations And Societal Impact:**

Limitations are addressed and potential negative societal impact is well described.

**Main Review:**

The image formation model is a valuable extension of (dynamic) neural radiance fields to raw depth measurements.

The experiments show good static reconstruction quality from two views already. The dynamic reconstruction quality is similar to Video-NeRF (unpublished at the time of submission I believe) even though Video-NeRF uses additional regularization losses.

I almost missed the index.html file in the supplemental folder and suggest to mention it in the supplemental PDF or the main paper.

Extrinsics:

The novel view synthesis results seem to be along different camera paths (and resolutions) for different methods. The NSFF camera circles seem to be a bit larger than for ToRF, e.g. Cupboard has a much more aggressive path for NSFF than for ToRF. How are the extrinsics for novel view rendering made comparable between methods? Are quantitative results reported on these differing NVS paths?

Comparisons:

- Both NSFF and Video-NeRF originally estimate depth from RGB images during preprocessing and then use that estimated depth during optimization. If computationally feasible, evaluating Video-NeRF and NSFF on both recorded and their respectively own estimated depths would be nice. Measured depth might show other artifacts than estimated depth while measured depth might be more accurate than estimated depth, so it is not clear to me which one is fairer for comparisons.
- Video-NeRF shows similar quality to ToRF. Is that still the case when using estimated (instead of recorded) depth for Video-NeRF? I.e. are the regularization losses in Video-NeRF sufficient to make up for the lack of recorded depth?
- In general, it would be nice to show NVS from viewpoints that are more different than currently. The results show NVS that is very close to the input camera path. I understand that this is challenging for this type of method currently, but maybe geometry supervision helps? The reconstructions look like they are not really correct (disappearing hands, torn hair), so I presume that ToRF does not work well for such cases.
- Similarly, larger camera motion would be nice (in which case Colmap might also get enough baseline to estimate extrinsics). As long as NVS sticks relatively close to the input camera path similar to NSFF, the quality should be decent.


Summary:

Overall, the method is a non-trivial extension to dynamic NeRF models and is valuable. For the dynamic setting, I believe that clarifications regarding the extrinsics are necessary and that additional evaluations on NSFF and Video-NeRF would strengthen the paper and provide useful insight but are not strictly necessary for acceptance.

Minor points:

- Not necessary since it is a preprint, but there is another concurrent dynamic NeRF paper: Yilun Du, Yinan Zhang, Hong-Xing Yu, Joshua B. Tenenbaum, Jiajun Wu. "Neural Radiance Flow for 4D View Synthesis and Video Processing", arXiv:2011.13084.
- line 136: "component component"
- The DinoPear and Dishwasher depth videos of ToRF are in the wrong color coding.
- Is Eq. 6 or Eq. 7 used for the results? I.e. are multiple paths supported?

After rebuttal:

My points were addressed well enough. My main concern was the comparability of the extrinsics across methods, but I understand the difficulty in achieving that and I don't see a better option than what the authors are promising for the revision.

While not mentioned by the authors in their summary of changes, they also promised that "We will include larger baseline view synthesis results in our revised submission."

There are a few remaining downsides that keep this submission from being outstanding, especially important to me is the comment by reviewer SWuj: "However, the quality of the results is not fully convincing to me, and considering that the paper uses a more complex acquisition setup, I would expect better results than the ones shown in the paper."

Overall, however, I believe that the submission does rise from slightly below to slightly above the threshold for acceptance given its novelty and timeliness.

**Time Spent Reviewing:**

4

---

> ### Author Response · Authors · 2021-08-09
> **Response**
>
> ***
> ### Extrinsics for view synthesis sequences
>
> Thank you for your helpful feedback. Many of the points raised relate to exactly how to implement the comparisons between different methods. We stress that this task was made more difficult because, out of the box, NSFF produces results that are not directly comparable (metric vs. arbitrary scale), and because VideoNeRF’s pose estimation using COLMAP frequently produced noisy poses on our scenes. This raises the tricky question of how much we should adapt other methods to make comparisons possible, e.g., which poses to use, or whether we should manually tune scene scale variables.
>
> The NSFF code worked when using COLMAP extrinsics. Even though the poses have some error, the method is robust in its ability to produce output thanks to the learning-based single-image depth estimation, which does not rely strongly on the recovered poses and mismatches in depth scale. As such, we did not need to adapt it for comparison.
>
> In contrast, VideoNeRF suffered when using COLMAP extrinsics. As VideoNeRF requires extrinsics and depth maps that are scale-consistent (see Section 4.3), we were forced to train VideoNeRF using extrinsics optimized via our method TöRF – without this, no comparison was possible. As such, this adapted VideoNeRF comparison is closer to “TöRF with derived depth” rather than with phasor images, i.e. Eq. 7 vs. Eq. 6 in our paper.
>
>
> **Depths:**
> For VideoNeRF, our current result _is_ VideoNeRF with measured ToF-derived depth. We expect that other depth estimation approaches that rely on COLMAP poses will perform worse (hence our adjustments). We will include a comparison with estimated depth to show this.
>
> For NSFF, we will add comparisons using our measured ToF-derived depth. We expect them to be better than the current results.
>
> **Camera paths:**
> We generated the spirals using each codebase’s built-in functions for path generation (TöRF, VideoNeRF, NSFF), and made these paths comparable by correcting for the relative scale of the COLMAP and TöRF extrinsics. This worked well for most scenes, but suffered where the COLMAP extrinsics were inaccurate. Note also that the monocular depth maps estimated in NSFF are inherently scale ambiguous, leading to differences in the camera paths relative to scene content – we will manually tune the spiral paths for NSFF to resolve these issues.
>
> **Camera resolutions:**
> This was a bug; as in our response to Reviewer rBB3, we accidentally trained VideoNeRF with higher-resolution input than all other methods. Our results are comparable in sharpness when using equal input resolution. See the updated Figure 5:
>
> * https://drive.google.com/file/d/11AiEzamLifmttTIzOMQJivm2WRVIerW7/view (anonymous link)
>
> **Quantitative results:**
> The BRISQUE quantitative results are reported for the paths shown in the supplement. The DinoPear results are reported for ground-truth held-out views that are the same for all methods.
>
> Finally, we would just like to reiterate that, beyond these comparisons, our claimed contribution is in using phasor images through a different image formation model and loss. This is a fundamental technical contribution that helps overcome the limitations of existing methods, as elaborated in our response to Reviewer rBB3.
>
> ***
> ### Additional baselines
>
> We will include a result comparing VideoNeRF and NSFF on measured and estimated depths.  As discussed above, we expect that VideoNeRF will perform worse with estimated depth.
>
> ***
> ### The reconstructions look like they are not really correct (disappearing hands, torn hair), so I presume that TöRF does not work well for such cases.
>
> As mentioned in Section 6.1, low-reflectance dynamic regions are challenging for our method. For the most part, our reconstructions are more correct than baselines, as shown in our supplemental video results.
>
> ***
> ### Larger camera motion for view synthesis
>
> We will include larger baseline view synthesis results in our revised submission.
>
> ***
> ### Larger camera motion in input data
>
> We agree that larger input camera motions would help COLMAP and thus potentially improve the results of other methods. We expect our approach not to suffer with larger motion given that our measured depth provides more accurate pose optimization in the more challenging case of smaller camera motion.
>
> ***
> ### Is Eq. 6 or Eq. 7 used for the results? I.e. are multiple paths supported?
>
> Equation 6 is used by our method; we will clarify this. Multiple paths are supported in the sense that a ray integrates multiple phasors, each of which is due to a different light path. This can also model multi-bounce light paths, e.g. reflections, by placing content at the depth/phase of the reflection. The PhoneBooth scene (reflection in glass) and the Cupboard scene (swinging door reflection in fridge) contain examples of dynamic reflections which are reproduced plausibly in view synthesis results.
>
>
> ***
> ### Neural Radiance Flow for 4D View Synthesis and Video Processing
>
> Thanks for the reference.
>
> ***
> ### The DinoPear and Dishwasher depth videos of TöRF are in the wrong color coding.
>
> Our mistake; this is now fixed.
>
> ***
> ### index.html in the supplemental material
>
> We will mention it in the PDF. We will also release a public webpage for the paper with all results, datasets, and code.

---

### Official Review · Reviewer_XGQC · 2021-07-16

**Rating:** 7
**Confidence:** 3

**Summary:**

The authors introduce a novel neural representation based on an image formation model for continuous-wave ToF cameras, thereby replacing the data-driven priors used by other approaches on the reconstruction of dynamic scene parts with measurements from a time-of-flight (ToF) camera. Using the modeling of raw ToF measurements instead of depth maps allows better handling challenging scenarios such as regions with low reflectance, multi-path interference, or the limited unambiguous depth range of sensors.


The proposed approach exhibits a potential benefit regarding robust static scene reconstruction from only a few views and regarding increased robustness of dynamic scene reconstruction. Still, the evaluation is lacking in some regards as discussed below.


**Limitations And Societal Impact:**

The authors provide a discussion of limitations and the potential societal impact.

**Main Review:**

Originality:
The proposed approach seems novel. There seems to be not much work yet on Time-of-Flight Radiance Fields. Related work seems to be mostly covered. I would recommend adding the recent work by Shen et al., Non-line-of-sight Imaging via Neural Transient Fields’ as it also considers time-resolved data.

Quality:
- Besides a discussion of the used hardware, the metrics used for evaluation and the used data the authors provide results for static and dynamic scenes.
- For static scenes the authors demonstrate in quantitative and qualitative comparisons that fewer views can be used in comparison to NeRF. Detailed experiments on different configurations/poses of the views have not been conducted.
- For dynamic scenes the authors provide quantitative and qualitative comparisons to NSFF and VideoNeRF, where the proposed method also seems to allow more plausible results in comparison to the other techniques. However, there seems to be a clear shift in the overall color distribution (Figure 5) of the depth which might indicate an offset or scale issue. This needs proper evaluation to allow getting insights on the real potential of the approach (I would also consider raising my voting by 1 point). Highlighting regions with zoom-ins could be another option to demonstrate the benefits for fine-grained structures.

Clarity:
- The paper is well-structured and easy to follow. Figures/tables and captions are informative.
- The introduction/motivation is good.
- The supplemental provides a comprehensive overview on different aspects and additional results.

Significance:
- The paper seems reproducible by more experienced readers. The authors mention that they aim to release materials for reproduction later on, but this seems not guaranteed.


Post-Rebuttal Comments:

The authors addressed concerns regarding ...
- the their depth map results (Figure 5) where a the color shift bug and a resolution bug have been resolved,
- additional zoom-ins to improve the exposition of the results,
- the influence of the number of view configurations.
I also appreciate the additional results they provided on other reviewer comments, e.g. regarding the effect of the accuracy of the depth quality for VideoNeRF and the effect of the number of samples used for rendering.

The authors' promise that "We will include larger baseline view synthesis results in our revised submission." (also highlighted by Reviewer EBPE) is indeed interesting, but at the moment still kept vague, i.e., they did not yet provide details regarding the behavior in this regards.

As a result of the clarification of bugs and the extension of the evaluation, I also see the submission now above the threshold for acceptance.



**Time Spent Reviewing:**

4

---

> ### Author Response · Authors · 2021-08-09
> **Response**
>
> ***
> ### Additional evaluation of static scene reconstruction
>
> As our focus is on dynamic scenes with a single RGB and ToF camera pair, a rigorous evaluation of different camera configurations is beyond the scope of this work. We found that, as we increase the number of views, NeRF yields slightly better RGB view synthesis results, with TöRF still outperforming NeRF in terms of depth error metrics. To show this, we will extend our 2 and 4 view configurations (Figure 4) with 8 and 16 views.
>
> ***
> ### Shift in the overall color distribution (Figure 5) of the depth maps
>
> The inconsistency in depth color maps for Figure 5 was a visualization bug – caused by inconsistent color map normalization between different methods and sequences. We had fixed this in the supplemental videos and document, which are correctly color-mapped. Please see these, plus our updated Figure 5 with zoom-ins at this anonymous URL:
> * https://drive.google.com/file/d/11AiEzamLifmttTIzOMQJivm2WRVIerW7/view (anonymous link)
>
> We are confident that there is not an offset or scaling issue in training, as this would result in clear artifacts for both TöRF and VideoNeRF: static content would appear to move, and view synthesis trajectories would not be as smooth. The reconstructions for NSFF are not metric, which makes it difficult to pick color mappings, but again in the supplemental videos all colormaps are relatively consistent.
>
> Please also see our response to Reviewer EBPE, as we will add new supplemental details describing precisely how we produce consistent color mappings across TöRF, NSFF, and VideoNeRF.
>
> ***
> ### Reproducibility
>
> We promise that we will release code for our method and simulator, as well as synthetic and real-world datasets. There are no copyright or intellectual property restrictions that will block the release of these items, as supported by our asset declaration in the NeurIPS Checklist and our supplemental material.
>
> ***
> ### Missing reference
>
> Thank you, we will add Shen et al.’s work.

---

### Official Review · Reviewer_rBB3 · 2021-07-18

**Rating:** 7
**Confidence:** 4

**Summary:**

This paper is trying to render novel views of static/dynamic scenes by using RGB and ToF sensors. Rather than directly using depth from ToF sensors, this paper takes a step further to work on phase images, and is more likely to handle challenges of using ToF sensors. Experiment results show that the rendered images look more realistic, with less artifacts, especially for dynamic scenes.

**Ethical Concerns:**

No.

**Limitations And Societal Impact:**

It's a nice paper that might interest many people in CV and graphics. The multi-resolution issue should be addressed for better visual quality. Also, the rendering speed should be accelerated. Both might be the main hurdles toward real applications.

**Main Review:**

1. There are quite a lot endeavors trying to extend NeRF to dynamic scenes, which is higly illposed. To merge depth information from a ToF sensor is a nice choice, and this paper seems to be the first in utilizing TOF in the NeRF framework. Furthermore, this paper further works on the phase images, thus is more likely to avoid some commen issues of ToF, in the presence of complex reflectance and depth noise.

2. The rendering model for ToF images has been carefully adapted in the NeRF framework, and those equations look correct to me.

3. As for the results, clear improvement can be observed, when compared with existing methods in which depth information is not used. This is reasonable, since more information is used here. However, the contribution of using phase ToF images, rather than depth directly generated from a ToF sensor, is not clear. Comparison should be added to show the difference, so as to justify the necessity of working from phase images (which is the core claim of this paper).

4. When comparing the results from this paper and those from NeRF, we can see that, the resolution and the sharpness of the proposed method are inferior to those from NeRF. One key reason might be the resolution difference between RGB and ToF. As we know, the rgb camera has high resolution, and the ToF sensor usually has low resolution. So, for a light ray in the ToF framework, there must be multiple corresponding light rays in the RGB framework.  This paper ignores this critical fact. It is highly recommended to examine this multi-resolution fact when using RGB and ToF bi-modality input.

5. As the authors admitted, dark regions indeed have obvious artifacts, which is likely to be enhanced by using some supervised signal enhancement techniques. This can be an interesting task in the future.

**Time Spent Reviewing:**

6

---

> ### Author Response · Authors · 2021-08-09
> **Response**
>
> ***
> ### The contribution of using phase ToF images, rather than depth directly generated from a ToF sensor, is not clear.
>
> Please see our supplemental web page for the clearest description of the benefits of ToF, where we show direct examples that using phasor images over depth provides: (i) robustness to phase wrapping, (ii) robustness to high phase noise in static, low-reflectance regions, (iii) the ability to model measurements that are the result of a mixture of phasors from different light paths.
>
> For (i), our results on the Photocopier sequence and the DinoPear sequence both incorporate ToF measurements with wrapped phase and show better results than VideoNeRF.
> Please see the updated quantitative results for DinoPear with wrapped and manually unwrapped depth (in reviewer SWuj), and corresponding qualitative result at this anonymous Google Drive link:
>
> * https://drive.google.com/drive/folders/1PjKa6kHysRufOiWyIpDxrPgqbPE73vTB (anonymous link)
>
> For (ii), on the DeskBox and Cupboard sequences – the monitor in DeskBox and the coffee machine in Cupboard have noisy depth/phase, which is resolved correctly through our approach, while this noise is reproduced in the VideoNeRF results. Note that *dynamic* objects with low reflectance remain a challenge.
>
> For (iii), see the PhoneBooth sequence: the reflections in the glass produce floating content in the VideoNeRF result, while our ability to model measurements as a mixture of phasors produces a more plausible view synthesis sequence.
>
> We will revise the text to further emphasize these comparisons.
>
> ***
> ### RGB and ToF resolution
>
> Thank you for these points. We must clarify an important point: for our real sequences, the resolution of the RGB and ToF images are both 320×240 pixels; we downscale the input RGB images by two (for faster training). Thus, our results are not low-resolution wrt. NeRF due to differing RGB and ToF resolutions; simply that we train on lower-resolution inputs. That said, the point is well taken: to handle high-resolution color and low-resolution ToF, it is in principle possible to adapt an approach similar to Mip-NeRF [b] and trace larger ray bundles/cones for the ToF images. Note that our current approach of reducing the weight on ToF data over time can aid in supporting higher resolution RGB images (see Section 4.2). We will add new discussion to consider these points.
>
> [b] Mip-NeRF: A Multiscale Representation for Anti-Aliasing Neural Radiance Fields, Barron et al., arXiv 2021.
>
> ***
> ### Limitations in dark regions and rendering speed
>
> We look forward to helping our community try to solve these problems in the future!

---

### Official Review · Reviewer_SWuj · 2021-07-18

**Rating:** 7
**Confidence:** 4

**Summary:**

The paper proposes a method to reconstruct dynamic radiance fields from   a single monocular video for novel view synthesis. Different from previous
methods that rely on RGB images only, the paper makes use of both RGB and
time-of-flight cameras and use the phasor information of the ray to supervise
the geometry reconstruction. They derive the image formation model for the
ToF imagee. The experiment results show that the proposed method generates better depth estimation and dynamic scene view synthesis results.


**Limitations And Societal Impact:**

Please see the main review.

**Main Review:**


### Strengths
1. The usage of ToF camera for dynamic scene reconstruction is interesting
and novel. The paper shows improvement over previous method that replies on
RGB or RGB+depth.

2. The proposed method generates more accurate depth estimations.

3. The paper shows results on diverse synthetic and real scenes to validate
the proposed method and design choices.

4. The method can generate better view synthesis results than baseline methods
on sparse inputs.

### Weakness

1. While the paper shows that the proposed method achieves better BRISQUE scores,
for most of the video results, the results of the proposed method are
more blurry, and the results of VideoNeRF has more fine-grained details. What is the
reason behind this? Also the proposed method has clear artifacts around depth
discontinuities and obvious distortions (StudyBook, Dishwasher). While the paper
acknowledges such limitations, they weaken the contributions of the paper.

2. BRISQUE is not a common metric used for this task. I would suggest the paper should
have comparison to ground truth on real scenes by capturing multi-view RGB images simultaneously and use some of the views as testing and report the standard metrics such as PSNR, SSIM
and LPIPS. Currently the paper compares to VideoNeRF on synthetic data (Table 2 in supp)
and use the GT depth for it, and from the results we can see that VideoNeRF is better than
TORF consistently. However, we cannot draw the conclusion whether VideoNeRF is still better
when using the ToF depth. Therefore, I would suggest adding quantitative comparisons on
real data with GT.

3. The infrared camera is usually sensitive to background light and sunlight. It's not
clear to me whether the method works well for outdoor scenes and scenes with visible
light sources. Is the method robust the potential corruptions in the ToF information?

4. The synthetic data such as Bedroom and bathroom has a lot of noise. It would be better
to increase the number of samples during rendering and generate noise-free images.

Overall I like the idea of using ToF cameras to provide additional supervision signals of
neural radiance fields, and the paper is able to show improvement over previous methods.
However, the quality of the results is not fully convincing to me, and considering that
the paper uses a more complex acquisition setup, I would expect better results than the
ones shown in the paper. Therefore I vote for "marginally above the acceptance threshold".



**Time Spent Reviewing:**

2.5

---

> ### Author Response · Authors · 2021-08-09
> **Response**
>
> ***
> ### Why are the results of TöRF more blurry than VideoNeRF?
>
> For faster training, we downscaled the input RGB images by a factor of 2 (e.g., from 640×480 to 320×240 for real-world scenes, and from 512×512 to 256×256 for synthetic scenes).  However, we inadvertently trained VideoNeRF on higher resolution input images, resulting in sharper results and better quantitative metrics in the output.
>
> We provide an updated Figure 5, with all methods trained at the same resolution:
>
> * https://drive.google.com/file/d/11AiEzamLifmttTIzOMQJivm2WRVIerW7/view  (anonymous link)
>
> When trained on the same resolution input images, sharpness is approximately the same for all methods.
>
> ***
> ### Paper should compare to ground truth on real scenes by capturing multi-view RGB images simultaneously.
>
> We decided to use synthetic RGBD data to quantify performance for both NVS images **and** depth maps, whereas capturing real-world multi-view RGB images would only allow us to evaluate NVS images. For this, we adapted a physically-accurate path tracer to evaluate methods with hold-out reference views and ground truth depth, reporting PSNR, SSIM, and LPIPS numbers.  We believe that this still provides a fair quantitative evaluation of the methods.  We will release the path tracer for generating more synthetic datasets.
>
> ***
> ### Paper compares to VideoNeRF on synthetic data, which uses GT depth as input
>
> Below, we provide the updated quantitative results for VideoNeRF trained with the lower resolution input images, and with **ToF-derived depth** rather than ground-truth depth. Since the ToF-derived depth contains wrapping ambiguities, we also manually unwrap depth (by adding 2π to all phase values below a certain threshold). We outperform VideoNeRF on all metrics for both wrapped and unwrapped ToF-derived depth.
>
> | Method | MSE (D) ▼ | PSNR ▲ | SSIM ▲ | LPIPS ▼ |
> |---|---|---|---|---|
> | VideoNeRF | 0.011±0.002 | 19.75±1.07 | 0.275±0.021 | 0.041±0.016 |
> | VideoNeRF (manually unwrapped) | 0.009±0.002 | 20.72±1.03 | 0.350±0.033 | 0.032±0.016 |
> | TöRF | 0.005±0.001 | 22.19±1.75 | 0.561±0.052 | 0.028±0.011 |
> |
>
> Please refer to the following Google Drive link for the corresponding updated video results:
>
> * https://drive.google.com/drive/folders/1PjKa6kHysRufOiWyIpDxrPgqbPE73vTB (anonymous link)
>
> ***
> ### The infrared camera is usually sensitive to background light and sunlight. It's not clear to me whether the method works well for outdoor scenes and scenes with visible light sources. Is the method robust the potential corruptions in the ToF information?
>
> Yes - the method is robust to corruption of the ToF measurements through the RGB signal.
>
> The reviewer correctly points out that the operating range of C-ToF cameras is lower outdoors and under strong ambient light.  While a C-ToF camera does remove ambient signal at the pixel level (therefore making it insensitive to ambient light [a]), the Poisson noise of the ambient signal remains.  This results in noisier phasor images, effectively reducing the range of the C-ToF camera.
>
> While all results were captured indoors, noise affects dark objects (see Supplemental Figure 4), objects placed further away (see the far wall in the Photocopier sequence), and even objects lit by strong ambient sources.  For example, in the PhoneBooth sequence, strong outdoor light is reflected off of the glass window.  In all of these cases, TöRF relies less on the phase of the ToF measurements and more on triangulation cues.
>
> Note that hardware solutions do exist for extending ToF range, including EpiToF (50 metres) [1] and scanning LiDAR systems.
>
> [1] Achar et al., “​​Epipolar Time-of-Flight Imaging”
> [a] Callenberg et al., “Snapshot Difference Imaging using Time-of-Flight Sensors”
>
> ***
> ### Increase the number of samples during rendering to reduce noise.
>
> We agree and will re-render these scenes.  The synthetic RGB and phasor images are both affected by render noise, which will be reduced by using a larger number of samples.  Sampling noise affects all of the methods within our comparison, so we do not believe that reducing noise would affect the conclusions of our analysis.
>
> ***
> ### Quality of results given the more complex acquisition setup.
>
> TöRF is inspired by new devices like the latest iPhone and iPad, as ToF data is likely to be generally available to exploit.  We built our imaging setup to access raw RGB and ToF images because raw ToF is currently not accessible on iOS devices.
>
> Concerning what quality we should expect, we mention two points:
> 1. Our formulation to incorporate raw ToF phasor measurements into NeRF overcomes the limits of naive ToF depth integration (e.g., see Figure 3; “benefits” section of supplemental webpage), such as correcting phase wrapping and handling objects with low IR reflectance. This provides better depth than is possible from ToF sensors or RGB sensors alone, supporting improved pose estimation, reconstruction, and NVS.
> 2. Results from a two-sensor RGB+ToF system are effectively always _extrapolating_ for dynamic objects, which limits quality. This is in contrast to bulkier multi-camera setups that pose an interpolation problem for dynamic scene view synthesis [6; 46 cameras] and [19; 17 cameras].
>
> [6] Immersive Lightfield Video, Broxton et al., SIGGRAPH 2020.
> [19] Neural 3D Video Synthesis, Li et al., arXiv 2021.

---

> > ### Comment · Reviewer_SWuj · 2021-09-02
> > **Reviewer feedback**
> >
> > I thank the authors for the rebuttal. It addresses my concerns. Overall I like the idea of the paper and I think the formulation of ToF in neural rendering is novel. Therefore, I would increase my score to 7.

---

### Author Response · Authors · 2021-08-09
**Summary of changes**

We thank all reviewers for their constructive and thoughtful comments. We will address reviewer concerns in the final version of the manuscript with the following changes:

* We will add a result that shows NSFF and VideoNeRF performance with both measured depth (from ToF camera) and estimated depth (using their respective depth estimation methods);
* We will re-render synthetic scenes with more samples-per-pixel to reduce render noise;
* We will manually tweak NVS camera paths to be more consistent across methods;
* We will fix typos, add the missing references, and correct visualization issues.

In addition, we have already made the following changes:
* We updated Figure 5 so that all methods use the same input RGB resolution;
* We updated synthetic VideoNeRF results to use ToF-derived depth instead of ground-truth depth.

Please see responses below for more details regarding all of these changes.

Finally, we will release all data and code associated with TöRF, including the ToF path tracer, to ensure that the method is fully reproducible.

With best wishes,

_The authors_

---

### Decision · Program_Chairs · 2021-09-27

**Decision:**

Accept (Poster)

**Comment:**

The paper proposed an approach to model geometry and appearance in a dynamic scene based on monocular RGB and ToF measurements. The key novelty is a principled extension of NeRF based on the image formation model of ToF measurements. The paper initially received a mixed rating, with two reviewers rated it below the bar and two above. Two reviewers were satisfied with the rebuttal and upgraded the ratings. While the meta-reviewer agrees that the quality of the results was not impressive, the ToF extension of NeRF for dynamic scenes is a solid technical contribution. The authors are encouraged to incorporate the review feedback in the final manuscript.